# Examining the Associations between Psychological Flexibility, Mindfulness, Psychosomatic Functioning, and Anxiety during the COVID-19 Pandemic: A Path Analysis

**DOI:** 10.3390/ijerph17238764

**Published:** 2020-11-25

**Authors:** Benita Wielgus, Witold Urban, Aleksandra Patriak, Łukasz Cichocki

**Affiliations:** 1The Education of Research and Development Center, Babinski Clinical Hospital, 30-393 Cracow, Poland; 2Department of Psychiatry, Rydygier Specialist Hospital, 31-826 Cracow, Poland; urwit247@gmail.com; 3Faculty of Philosophy, Institute of Psychology, Jagiellonian University, 30-060 Cracow, Poland; a.patriak@gmail.com; 4Department of Psychiatry, Andrzej Frycz Modrzewski Cracow Academy, 30-705 Cracow, Poland; lwcichocki@gmail.com

**Keywords:** coronavirus disease (COVID-19), COVID-19-related stress, anxiety, mindfulness, psychological flexibility

## Abstract

Social distancing plays a leading role in controlling the spread of coronavirus. However, prolonged lockdown can lead to negative consequences in terms of mental health. The goal of the research is to examine the relationship between anxiety and general psychosomatic functioning during the COVID-19 pandemic; the impact of psychological flexibility and mindfulness is also considered. Variables were measured with self-report questionnaires and symptom checklists. The sample included 170 people (M = 27.79, SD = 8.16). Pearson’s correlation, stepwise regression, and path analysis were conducted. The results showed a significant positive relationship between state anxiety and somatic and psychological responses to the pandemic. Path analysis revealed that mindfulness had a direct negative impact on and decreased the level of state anxiety (b = −0.22, *p* = 0.002), whereas psychological flexibility influenced the variable indirectly (b = 0.23, *p* = 0.002) by enhancing psychosomatic functioning (b = −0.64, *p* < 0.001). Psychological flexibility and mindfulness may mediate the development of mental disorders and facilitate achieving overall wellbeing. The study points to the usefulness of mindfulness practice as a form of self-help with anxiety symptoms; this is crucial during the pandemic because contact with clients is restricted.

## 1. Introduction

In December 2019, an outbreak of a novel coronavirus pneumonia was reported in Wuhan (Hubei, China). The virus was named severe acute respiratory syndrome coronavirus 2 (SARS-CoV-2) [1,2]. As of June 2020, this novel coronavirus had spread across the globe, affecting over 10 million people and causing more than half a million deaths [3]. The course of the infection caused by SARS-CoV-2 is varied from asymptomatic to severe respiratory distress and death. However, at this stage, symptomatic treatment is the main therapeutic choice [4,5].

The development of a COVID-19 vaccine is currently in progress, and an effective cure is still unknown [6]. Meanwhile, isolation remains effective protection against the virus [7]. Social distancing and quarantine are both recommended by the World Health Organization (WHO) [8]. On the other hand, social distancing has been proven to trigger negative mental health consequences, including intensified anxiety and depression [9,10] and/or apathy and hypersensitivity; it is especially worrisome that it can also exacerbate mental illness [11,12]. Furthermore, the reduction of interpersonal contact has a negative impact on wellbeing [13]. Additionally, information publicized by the media about exaggerated dangers can provoke misinformation, which results in decreased psychological functioning in the whole population [14]. On the other hand, mass and social media create a perceived sense of control and promote the prevention of erroneous practices in viewers by encouraging the use of safe health practices [15,16].

Recently, models based on personal resources and psychological resilience have been gaining popularity [17]. A growing number of studies have provided evidence that psychological flexibility is associated with one’s ability to cope in stressful situations [18]. This construct is defined as the ability to be in touch with oneself and with all present experiences (thoughts, feelings, and behaviors), as well as being able to act according to personal values even under unfavorable internal and external experiences [19]. Previous studies have shown that high psychological flexibility is associated with a better quality of life and other psychological outcomes [20]. The psychological flexibility model underlies acceptance and commitment therapy (ACT), which has satisfying empirical evidence for clinical outcomes in anxiety, depression, and PTSD [21]. The difference between ACT and traditional cognitive behavioral therapy (CBT) is that ACT tends to promote acceptance, while CBT focuses on changing maladaptive attitudes [20].

Mindfulness is classified as a method of emotional regulation. It is considered that the processes that occur during a mindfulness practice regulate difficult and unacceptable emotions, which, in turn, contribute to recovery. In this way, negative emotions are comprehended and can be transformed [22]. Studies concerning mindfulness and other relaxation techniques such as autogenic training or progressive training have demonstrated that meditation activates and increases the functionality of brain regions involved in learning and memory processing, perspective-taking, and self-referential processing; it also improves attention processes. In contrast, distress can contribute to reduced activation in these areas and in the amygdala and hippocampus [23]. Studies concerning mindfulness-based interventions, for instance, mindfulness-based stress reduction (MBSR) [24] or mindfulness-based cognitive therapy (MBCT) [25], have brought evidence that mindfulness techniques can reduce symptoms of both anxiety [26,27] and depression [28,29] that are related to somatic symptoms [30] such as cardiovascular disease [31] or cancer [32].

Considering the pandemic conditions and the difficulty in getting professional help, developing mindfulness practices via the internet is especially interesting. Mindfulness meditation apps have low or medium effects on perceived stress or anxiety symptoms, although there is no evidence of their long-term effectiveness [33]. Other online mindfulness-based interventions are also promising, with the most promising findings for stress [34]. Despite the worse outcomes of online interventions when compared to face-to-face [35] alternatives, these techniques have the potential to strengthen coping mechanisms, especially in this uncertain and burdening time of the pandemic.

During this pandemic, many people suffer from prolonged social distancing, isolation, and separation from social support. Getting professional psychological or psychiatric help has become more difficult. Therefore, it is necessary to find alternative forms of therapy or self-help that can contribute to a reduction in the negative consequences of this crisis by decreasing uncertainty, anxiety, and other forms of psychical distress. In Poland, the first case of coronavirus was registered on 4 March 2020. The dynamics of coronavirus spread were moderate at the time that the research was conducted: 2554 cases on 1 April 2020, rising to 23,786 cases on 31 May 2020.

The main goal of the study is to examine the relationship between anxiety and general psychosomatic functioning during the COVID-19 pandemic, taking into account the influence of indirect factors such as psychological flexibility and mindfulness. Three hypotheses were formulated. First, it is anticipated that worse somatic and psychological responses to the pandemic will be related to higher levels of state anxiety. Second, it is expected that a lower level of the anxiety trait, a higher level of psychological flexibility, and a higher level of mindfulness will be associated with lower intensity of psychopathological symptoms. Third, worse psychosomatic functioning symptoms will increase the level of state anxiety. Furthermore, based on observations, a path model will be constructed for a graphical presentation of the interrelationship between trait anxiety, psychological flexibility, general psychosomatic functioning, mindfulness, and state anxiety.

## 2. Materials and Methods

### 2.1. Participants

The sample contained 170 people. The average age of the participants was 27 years. The demographics are shown in Table 1. Participants were recruited from the Facebook platform using so-called snowball sampling. The survey, participation agreement, and questionnaires were shared via a hyperlink. Everyone with this link could participate in the study, but young people were more likely to fill in the survey as this service is mostly used by younger generations. The responses and the following criteria were included in the analysis: male or female gender, age above 18 years old, and consent to participation in the study. Data from respondents below the age of 18 years old were excluded. There was no information about the number of participants that failed to complete the form as they could withdraw without any consequence at any time. The incomplete questionnaires were not counted nor saved. Participants’ data were stored according to Google’s privacy policy and transferred to a Microsoft Excel Sheet.

### 2.2. Psychological Measures

The State–Trait Anxiety Inventory (STAI) [36] is a standard tool used in many studies concerning anxiety [37]. The STAI comprises two separate self-reported scales: X1 for state anxiety and X2 for trait anxiety. Anxiety is usually described as a feeling of tension, unrest, or apprehension, and overall increased activity of the autonomic nervous system. The anxiety trait is an individual stable quality with differences in frequency and intensity. State anxiety fluctuates based on individual overlapping stressors. The level of state anxiety should be low in nonstressful situations and high when there is a perceived danger or threat [36]. Each scale comprises 20 items with a four-point Likert scale from 1 = “almost never” to 4 = “almost always”. The Polish version of the STAI showed good internal consistency with Cronbach’s alpha coefficients, ranging from 0.74 to 0.92 [38]. In this study, higher scores reflect higher levels of trait and state anxiety.

The Acceptance and Action Questionnaire–II (AAQ-II) [19] is the second version of a tool that was constructed based on the human distress model. It assumes that people are able to find different opportunities in critical situations by being in contact with the present moment, including with one’s thoughts and feelings. This model considers psychological flexibility as a mediator of change and as the main instrument of interventions, particularly in ACT. The scale consists of 7 items, rated on a 7-point Likert scale from 1 = “never true” to 7 = “always true”. The Polish adaptation’s Cronbach’s alpha reliability has been reported as very satisfactory, scoring about 0.94 [39]. In this study, the internal consistency was also very good (α = 0.90). AAQ-II’s brevity and satisfactory psychometric properties seem to favor its use for online research. Higher scores mean better psychological flexibility.

The short version of the Freiburg Mindfulness Inventory (FIU-14) [40] is one of several instruments for measuring mindfulness. It can be used to evaluate the mindfulness of people who are not familiar with Buddhist teachings or other practices based on this concept, which is desirable in terms of our study where most of the participants were not familiar with mindfulness techniques. Mindfulness is described as a personal trait of being mindful of the moment-by-moment experience. Some of the features of mindfulness include openness, receptiveness, curiosity, and a nonjudgmental attitude. It assumes that an open and attentive individual is more in touch with feelings and experiences [41]. Fourteen items of the questionnaire define mindfulness as a process of regulating attention. The items are rated on a 4-point Likert scale, from 1 = “rarely” to 4 = “almost always”. The reliability measure of this short version is acceptable (Cronbach’s alpha between 0.79 and 0.86). The Polish version scored 0.75 [42]. In this study, Cronbach’s alpha was 0.80, which allows the results to be interpreted. Higher scores reflect higher levels of mindfulness.

The General Functioning Questionnaire (GFQ-58) [43] is a tool that is used for the assessment of the intensity of psychopathological symptoms based on overall functioning. The instrument consists of 13 scales that explore many aspects of psychosocial functioning and symptom severity: worse social functioning, lack of entertainment, bad interpersonal relationships, and clinical symptoms such as cognition disorders, anxiety disorders, addictions, depression, manic disorders, food disorders, sleep problems, sexual dysfunctions, and somatic symptoms. Participants rate items on a 5-point Likert scale (0 = “never” to 4 = “almost always”). The measure includes 58 items. As has been confirmed in several trials, the scale has good internal consistency, scoring between 0.89 and 0.92. In this research, GFQ-58 results had a Cronbach α coefficient of 0.94. GFQ-58 provides a variety of different scales that measure reduced psychosocial functioning, lack of entertainment, and bad interpersonal relationships, all of which seem to be especially important in relation to the COVID-19 pandemic. Moreover, its good psychometric properties and brevity make this questionnaire favorable for online research. Higher scores indicate better general functioning. This variable is referred to as psychosomatic functioning in the research.

To assess the intensity of stress and anxiety among participants impacted by the COVID-19 pandemic, a checklist of clinical features was prepared. Participants were scored in two categories of symptoms: firstly, a set of psychological symptoms that was referred to as “psychological response to the pandemic” (intense distress about the pandemic, sleep problems, hypersomnia, intrusive and upsetting thoughts about the pandemic, tension, restlessness and fear, anxiety, nightmares, depressed mood, hypervigilance, and irritability or anger); secondly, somatic symptoms, referred to as “somatic response to the pandemic” (headache, dizziness, stomach ache, nausea, apathy, and diarrhea). The reliability of the psychological symptoms list was acceptable (α = 0.82), but the somatic symptoms list was insufficient (α = 0.63).

### 2.3. Procedure

The questionnaires were preceded by a short description and explanation of the aim and form of the study. Participation in the research was voluntary and anonymous. The Frycz-Modrzewski Academy Ethical Committee approved the study. All procedures were conducted by the committee’s ethical standards and in concordance with the Helsinki Declaration of 1975, as revised in 2000.

The survey had three steps. First, participants filled demographic questionnaires (with information about age, gender, level of education, marital status, and information of residence), STAI X2 (anxiety as a trait), and GFQ-58. Second, participants were invited to read a short note describing the pandemic in order to activate the distress related to it. The third step consisted of the checklist of clinical symptoms, STAI X1 (to assess anxiety as a state in the last 7 days of the lockdown), AAQ-II, and FIU-14. The survey was created as a Google document to limit interpersonal contact. The contents of the text are in Appendix A.

The study was conducted from the beginning of April to the end of May 2020.

## 3. Results

In this study, the statistical analysis of the data was run using SPSS Statistics for Windows (version 25). R Project for Windows (version 4.0.2) was used to test the structural equation models (SEMs). The first step was to identify the associations between the variables and perform a regression analysis. Based on the regression results, structural equation models were constructed and tested to determine the interrelationships between the variables.

Briefly, 170 participants (53.5%, *n* = 79) declared that during the last 7 days, they had felt at least one of the somatic symptoms on the checklist; 72.4% (*n* = 123) had been affected by at least one of the psychological symptoms caused by the pandemic. In the case of the somatic symptoms, the most often reported were headaches, which were reported by 40.6% (*n* = 69), apathy, 22.4% (*n* = 38), stomach ache, 12.9% (*n* = 22), and dizziness, 11.2% (*n* = 19). Less frequent were nausea, which was reported by 8.8% (*n* = 15) and diarrhea by 5.9% (*n* = 10). On the other hand, the most frequently reported psychological symptoms were depressed mood, reported by 44.7% (*n* = 42); tension, 34.1% (*n* = 58); sleep problems, 27.1% (*n* = 15); followed by restlessness and fear, reported by 24.7% (*n* = 37); hypersomnia, 23.5% (*n* = 40); distress, 22.4% (*n* = 28); irritability or anger, 21.8% (*n* = 37); anxiety, 21.2% (*n* = 36). The least common were intrusive and upsetting thoughts about the pandemic, which were reported by 15.3% (*n* = 26); nightmares, 13.5% (*n* = 26); hypervigilance, 5.9% (*n* = 10).

### 3.1. Correlations between Personal Resources and Indicators of Psychosomatic Functioning during the COVID-19 Pandemic

Pearson’s correlation analysis was used to explore the associations between anxiety (state and trait), psychological flexibility, mindfulness, and somatic and psychological responses as a maladaptive reaction to the pandemic. Table 2 shows the results.

There were significant positive correlations between somatic and psychological responses (r(168) = 0.578, *p* < 0.001) and between somatic (r(168) = 0.430, *p* < 0.001) and psychological responses (r(168) = 0.678, *p* < 0.001) with state anxiety. In addition, higher levels of somatic symptoms were associated with higher levels of trait anxiety (r(168) = 0.392, *p* < 0.001), lower levels of psychological flexibility (r(168) = −0.350, *p* < 0.001), lower levels of mindfulness (r(168) = −0.244, *p* < 0.001), and lower levels of psychosomatic functioning (r(168) = −0.505, *p* < 0.001). The higher the level of psychological symptoms, the higher the levels of trait anxiety (r(168) = 0.431, *p* < 0.001) and the lower the levels of psychological flexibility (r(168) = −0.300, *p* < 0.001), mindfulness (r(168) = −0.348, *p* < 0.001), and psychosomatic functioning (r(168) = −0.597, *p* < 0.001). There were significant negative correlations between state anxiety and psychological flexibility (r(168) = −0.370, *p* < 0.001), mindfulness (r(168) = −0.458, *p* < 0.001), and psychosomatic condition (r(168) = −0.659, *p* < 0.001). The correlation between state anxiety and trait anxiety was positive (r(168) = 0.549, *p* < 0.001). There were significant positive associations between psychosomatic functioning and psychological flexibility (r(168) = 0.613, *p* < 0.001) and mindfulness (r(168) = 0.502, *p* < 0.001).

### 3.2. Stepwise Regression for Psychosomatic Functioning

Regression analysis was used to evaluate significant predictors of general psychosomatic functioning. Anxiety trait, psychological flexibility, and mindfulness were entered into Model 1, as shown in Table 3. The variables accounted for 51% of the variance of psychosomatic functioning.

Next, regression analysis was used to assess psychosomatic functioning as a predictor of anxiety. As presented in Table 4, psychosomatic functioning explained 43.4% of the variance of state anxiety related to the pandemic.

### 3.3. A Path Model for Pandemic Anxiety

Following the correlation analysis and the two regression models, which confirmed indirect relationships between anxiety trait, psychological flexibility, mindfulness, and psychosomatic functioning, the cumulative impact of these variables on the state of anxiety in relation to the coronavirus pandemic was especially interesting. To measure the associations between these variables and explore their importance for anxiety in relation to the pandemic, path analysis was conducted using the structural equation model (SEM). All the goodness-of-fit indices had ideal fits (χ^2^/*df* = 2.91, df = 2, CFI = 0.998, TLI = 0.990, RMSEA = 0.052, SRMR = 0.014). The proposed model is presented in Figure 1.

The model explained 46.5% of the variance in state anxiety, 52.3% of the variance in psychosomatic functioning, 50.8% of the variance in psychological flexibility, and 46.3% of the variance in mindfulness. The results showed that psychosomatic functioning (*b* = −0.64, *p* < 0.001) and mindfulness (*b* = −0.22, *p* = 0.002) had a direct negative impact on state anxiety, but psychological flexibility did not (*b* = 0.14; *p* = 0.068). The direct path between psychological flexibility and psychosomatic functioning was positive and significant (*b* = 0.23, *p* = 0.002), but mindfulness did not signify psychological flexibility (*b* = 0.14, *p* = 0.065). Trait anxiety was significantly and negatively associated with psychosomatic functioning (*b* = −0.54, *p* < 0.001), psychological flexibility (*b* = −0.61, *p* < 0.001), and mindfulness (*b* = −0.68, *p* < 0.001). Table 5 shows the results.

## 4. Discussion

The main aim of the study is to evaluate the associations between anxiety and general psychosomatic functioning during the COVID-19 pandemic and to recognize the influence of indirect factors such as psychological flexibility and mindfulness on these variables. The study had a three-step procedure: first, basic demographic data were collected from the participants, who were then asked questions about the epidemic, trait anxiety, and general psychosocial functioning; secondly, participants read a short text concerning the pandemic that was intended to arouse distress. Next, the quantity of their clinical somatic and psychological symptoms was checked, along with their level of state anxiety, psychological flexibility, and mindfulness.

The first finding showed that worse somatic and psychological responses to the pandemic were associated with higher levels of state anxiety (first hypothesis) and worse general psychosomatic functioning. It was assumed that the state anxiety scores could be considered an adequate measure of pandemic anxiety and general wellbeing. These results have been confirmed by other studies that have shown that adults are more likely to feel anxious and stressed due to the COVID-19 pandemic [44,45]. A study examining the general population in China showed that 53.8% of participants reported a rise in psychological symptoms from moderate to severe, in particular anxiety, depressive symptoms, and stress [46]. Similar studies were conducted in Spain [47] and Australia [48], where 78% of participants declared that they had been feeling worse since the outbreak started. In Poland, there was a similar trend. In this study, young adults reported the occurrence of at least one somatic symptom (53.5%) and one psychological symptom (72.4%) during the week preceding the participation in the study. Headaches and apathy were reported as the most intensified compared to other somatic symptoms like stomach ache, dizziness, nausea, or diarrhea. Furthermore, the psychological symptoms most likely to occur were depressed mood, tension, and sleep problems, followed by restlessness and fear, hypersomnia, distress, anxiety, irritability, and anger. Less frequent were intrusive and upsetting thoughts about the pandemic, nightmares, and hypervigilance.

The correlation analysis results supported the second hypothesis regarding the associations between psychosomatic functioning and anxiety and psychological flexibility and mindfulness separately. Strong associations between general psychosomatic functioning and anxiety, psychological flexibility, and mindfulness were observed, as measured by both psychometric tools (GFQ-58) and the symptoms checklist. The regression indicated that psychological flexibility was a predictor of general psychosomatic functioning, but mindfulness was not. In this case, improving psychological flexibility and reducing anxiety favor better mental health outcomes, even in more stressful situations like a pandemic. Some studies have confirmed that enhancing personal resources guarantees a better response to unfavorable and unpredictable circumstances [49], including during the pandemic [50,51]. Nevertheless, the second hypothesis was not completely confirmed as mindfulness was not a predictor of general psychosocial functioning. Furthermore, the data indicates the significant importance of trait anxiety on psychological flexibility and mindfulness. Reportedly, high levels of trait anxiety significantly reduce psychological flexibility and mindfulness.

The path analysis model was constructed based on correlation and regression analyses in order to evaluate the moderating role of psychosomatic functioning (GFQ-58), psychological flexibility (AAQ-II), and mindfulness (FIU-14) on state anxiety (STAI), taking into account the influence of trait anxiety (STAI) on the mentioned variables. Path analysis revealed that psychological flexibility and mindfulness each had a different path impact on state anxiety. In the path analysis model, mindfulness directly reduced the level of state anxiety. At the same time, psychological flexibility promoted better psychosomatic functioning and indirectly diminished the level of state anxiety. Ipso facto, the third hypothesis has been confirmed.

It was somewhat surprising that mindfulness had an impact on state anxiety, but it did not predict general psychosomatic functioning in the regression model. The current results imply that mindfulness should be differentiated as a personal trait [52] and as an ability that is acquired as a result of practice [53,54]. The intensity of mindfulness was tested among young people who had not practiced it before. Hence, people who can be described as more mindful and open to different experiences are more able to reduce their anxiety. Nevertheless, mindfulness training favors better emotional regulation and has an impact on overall wellbeing and mental health. Practices based on mindfulness improve the quality of sleep [55] and of life [56]. Similarly, different relaxation techniques affect psychological outcomes by decreasing anxiety and depression symptoms [57,58].

This study builds upon previous ones as it explores the cumulative impact of psychological flexibility, mindfulness, and psychosomatic functioning on pandemic-related anxiety. The relationship between mindfulness and anxiety is direct, as was suggested by previous studies [59,60]. Mindfulness is a stable protector against pandemic-related distress [61]; it also reduces levels of anxiety, depression, and emotional exhaustion and improves harm avoidance, especially in the most vulnerable, less resilient groups [59]. These results are confirmed by the WHO, the Center for Disease Control and Prevention (CDC), and other organizations that promote mindfulness practices during the pandemic [8,62]. Additionally, prior studies suggest that psychological flexibility is a significant protective factor against the adverse effects of general and peritraumatic distress [63]. This study sheds new light on this notion as psychological flexibility appears to have an indirect relationship with pandemic-related anxiety by promoting better psychosomatic functioning. It seems that this link is more complicated than it first appeared to be. However, these conclusions are not undisputed, and the report requires further exploration.

This study has some limitations. First, the sample consisted of a homogeneous group of young and well-educated participants. The research established that mindfulness increased with age [64]. Moreover, with increasing age, people become more resilient to psychological distress, even during a pandemic [61]. Older people were underrepresented in this study; therefore, no further conclusions can be drawn. It is worth noting that women’s representation in the sample was higher than 70%. It is not quite clear why there was such a disproportion between men and women as the survey was held online and was easily accessible. One of the possible explanations is that women are more willing to partake in a study concerning their anxiety. Next, the research was conducted when coronavirus was spreading at a moderate rate; at that time, the restrictions in Poland were very strict. The reliability of the somatic symptoms list was insufficient (α = 0.63) and should be interpreted with caution. Moreover, there was no information about the state of the participants’ mental health before the pandemic. Additionally, it is important to remember that it is not clear whether the note in the second step of the survey actually enhanced state anxiety in participants. Its meaning was to activate the subject of the pandemic and to intensify the feelings and anxiety experienced during the lockdown in the week preceding participation in the study. However, high correlations between somatic and psychological symptoms caused by the pandemic and the measured level of state anxiety were sufficient to consider STAI X1 an appropriate measure of the intensity of state anxiety caused by the pandemic. Another significant limitation was the online survey, which made it impossible to control confounding variables. The study used snowball sampling, which is a nonrandomized sampling method that could bias the data. This procedure was recommended as safe for participants because of the dangerous epidemic conditions in Poland on 11 March 2020.

## 5. Conclusions

The results draw attention to the ways in which one can protect one’s mental health in special circumstances such as a pandemic. Psychological flexibility and mindfulness may mediate the development of mental disorders, protect against the unpleasant feelings caused by social distancing, and facilitate achieving overall wellbeing. Moreover, this study shows the usefulness of mindfulness techniques when coping with specific reactions to the COVID-19 pandemic and prolonged isolation. The primary advantage of mindfulness practice is that it can be applied without guidance to alleviate mild symptoms of stress, which is particularly important during epidemic restrictions. Mindfulness training can be used by medical staff (especially psychologists and psychotherapists) when direct contact with a patient or client is impossible or when access to more specific forms of help is too difficult.

## Figures and Tables

**Figure 1 ijerph-17-08764-f001:**
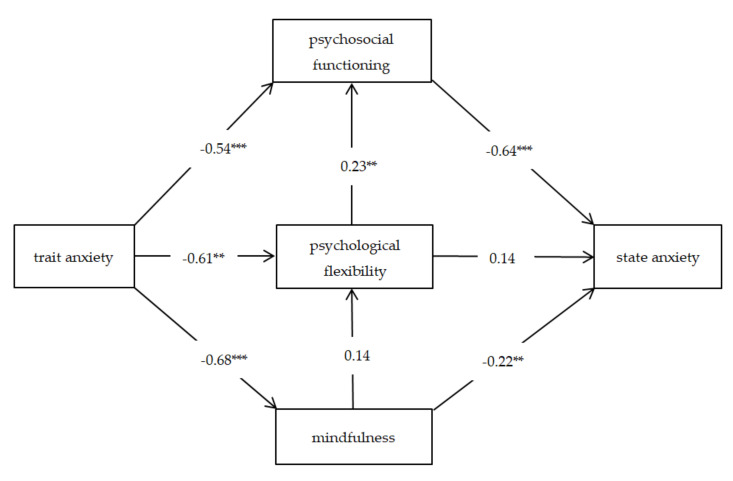
The path analysis model with standardized beta weighting for psychological flexibility, mindfulness, psychosomatic functioning, and anxiety (state and trait). Note: ** *p* < 0.01, *** *p* < 0.001.

**Table 1 ijerph-17-08764-t001:** Demographic characteristics of the sample (N = 170) and descriptive statistics for variables.

Demographic Characteristics	N	*M*	*SD*	Range	%
Age		27.79	8.157	18–59	
Gender					
Men	45				26.47
Women	125				73.53
Education					
Primary	8				4.71
Secondary	38				22.35
Postsecondary	21				12.35
Higher	103				60.59
Marital status					
Single	136				80.00
Married	32				18.80
Divorced	1				0.60
Widow/widower	1				0.60
Place of residence					
Village	24				27.90
Small town (less than 50 k)	29				72.10
Medium-sized town (from 50 to 150 k)	13				7.65
Big city (more than 150 k)	104				6.12

**Table 2 ijerph-17-08764-t002:** Pearson’s correlation between all variables.

Variables	1	2	3	4	5	6	*M*	*SD*
1. State anxiety	-						2.36	2.359
2. Trait anxiety	0.549 ***	-					2.23	2.229
3. Psychosomatic Functioning	−0.659 ***	−0.704 ***	-				3.93	3.926
4. Psychological Flexibility	−0.370 ***	−0.705 ***	0.613 ***	-			4.52	4.519
5. Mindfulness	−0.458 ***	−0.681 ***	0.502 ***	0.553 ***	-		2.47	2.473
6. Somatic response to pandemic	0.430 ***	0.392 ***	−0.505 ***	−0.350 ***	−0.244 ***	-	0.17	0.270
7. Psychological response to pandemic	0.678 ***	0.431 ***	−0.597 ***	−0.300 ***	−0.348 ***	0.578 ***	0.23	0.244

Note: *** *p* < 0.001.

**Table 3 ijerph-17-08764-t003:** Regression analysis model of anxiety trait, psychological flexibility, and mindfulness as predictors of psychosomatic functioning.

Model 1	F (3169)	R	R^2^_sk_	Predictors	Semipartial Correlation	% of Variance	*p*
General psychosomatic functioning	60.57	0.72	0.51	Anxiety trait	−0.329	10.8	<0.001
				Psychological flexibility	0.162	2.6	0.003
				Mindfulness	0.008	-	0.881

**Table 4 ijerph-17-08764-t004:** Regression analysis model of psychosomatic functioning as a predictor of state anxiety.

Model 2	F (3169)	R	R^2^_sk_	Predictors	Semipartial Correlation	% of Variance	*p*
Anxiety state	128.65	0.66	0.43	Psychosomatic Functioning	−0.659	43.4	<0.001

**Table 5 ijerph-17-08764-t005:** Regression analysis model of psychosomatic functioning as a predictor of state anxiety.

Path			Standardization Coefficient	*SE*	t	*p*	95% CI (Lower, Upper)
Psychosomatic functioning	→	State anxiety	−0.636	0.075	−8.698	<0.001	−0.652, −0.636
Psychological flexibility	→	State anxiety	0.139	0.026	1.826	0.068	0.048, 0.139
Mindfulness	→	State anxiety	−0.216	0.074	−3.115	0.002	−0.231, 0.216
Psychological flexibility	→	Psychosomatic functioning	0.232	0.025	3.103	0.002	0.077, 0.232
Trait anxiety	→	Psychosomatic functioning	−0.540	0.089	−7.225	<0.001	−0.647, −0.540
Mindfulness	→	Psychological flexibility	0.136	0.229	1.848	0.065	0.424, 0.136
Trait anxiety	→	Psychological flexibility	−0.613	0.263	−8.345	<0.001	−2.199, −0.613
Trait anxiety	→	Mindfulness	−0.681	0.065	−12.112	<0.001	−0.781, −0.681

Note: SE = standard error; CI = confidence interval.

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
