# Peer review of "Examining the Associations between Psychological Flexibility, Mindfulness, Psychosomatic Functioning, and Anxiety during the COVID-19 Pandemic: A Path Analysis"

_ijerph, 2020, doi:10.3390/ijerph17238764_

Round 1
Reviewer 1 Report
The manuscript investigated the associations between psychological flexibility, mindfulness, psychosomatic functioning, and anxiety during the actual COVID-19 pandemic by means of a path Analysis. Overall, the manuscript is well presented, understandable and it is supported by an appropriate research design. English language is appropriate in most part of the manuscript, however some other parts would benefit of major clarity and being revised due to their structure and grammar.
I would suggest therefore to accept the paper for publication after minor revisions. Among these however three points represent a major concern and I think must be revised before moving forward:
- Materials and Methods.:
- The Psychological measures should be introduced before the procedure. The current setting of the paragraph causes confusion as the reader must go up and down to understand which and when the tests were used.
- Moreover, the procedure section must be further extended. More details are needed in this section, the procedure section is too vague and needs more details.
- Line 106, “In Poland the first 1case of coronavirus was registered on the 4th of March. The dynamics of coronavirus spread were moderate during conducting the research with 2.554 cases on the 1st of April, reaching to 23.786 cases on the 31st of May”.
This may be more appropriate in the introduction.
- Line 102: Subsequently, they were invited to read a short note to activate feelings and distress connected with the pandemic:
The author states that the short note was used to activate feeling and distress connected with the pandemic, this was then measured employing the second part of the STAI.
My concern here is that the state anxiety measured after the short note that was provided to the participants is not measured against any baseline. It is not sure therefore the effect of the note on the state anxiety of the participants.
- Line 47 Coronavirus coverage in mass media creates beliefs of traumatic events in viewers which in turn deepens restlessness and panic [15].
Reference 15 is a paper that investigates exposure to TERRORIST ATTACKS. This does not seem to be a correct reference here. Even if the effect may be similar, it seems quite unreasonable to put on the same level the coverage of a Pandemic in mass-media with being exposed to terrorist attacks and its media coverage.
The following represent instead minor recommendations that the authors should take in consideration.
- Line 39. The process of creating a COVID-19 vaccine is currently in progress and the remedy is still unknown [6].
This sentence sounds a bit confusionary, vaccine seems to be considered a remedy(cure) for COVID19. This however is wrong as vaccine is not a remedy but instead is used to prevent the disease.
- Line 45 Additionally, information publicized by the media about exaggerated dangers can provoke mass trauma and result in decreased psychological functioning in the whole population [14].
This sentence seems to suggest that media are exaggerating the danger of the current pandemic provoking a mass trauma. The referenced paper however seems to point out something different. First, it investigates only social media, and not all type of media. Secondly, it suggests that despite social media can be useful during a Pandemic, they must be used carefully due to : “spending excessive time searching for COVID-19 news on social media given the infodemic and emotional contagion through online social networks.”
Line 74. During the pandemic many people can suffer from prolonged social distancing and isolation and remain in separation from social support. Getting professional psychological or psychiatric help becomes impossible.
This claim is not true, during the Pandemic, different local entities in different countries have provided professional psychological or psychiatric support to patients, therefore defining it IMPOSSIBLE is a really strong and false claim. I would suggest rephrasing this, stating that is more difficult.
Line 90 Sample included 170 young people.
This is not correct, age ranged from 18 to 58, while young adults generally range from 18 to 25
Line 98 to 109 full paragraph
This part contains few minor grammar issues that must be amended in order to facilitate the flow of the paragraph
Line 151 In this study, the statistical analysis of the data was tested using SPSS Statistics for Windows
Statistical analysis was run is more appropriate.
Linea 235 Those results find their confirmation in other studies where adults were more likely to feel anxious and stressed on account of the COVID-19 pandemic outbreak [39,40]. The study examining the general population in China showed the rise in psychological symptoms reported by 53.8% from moderate to severe levels, in particular concerning anxiety, depressive symptoms and stress [41].
Simple English style change, “the study examining ….” Would change with A study….. Otherwise it is not clear for the reader if the author is talking about a previous cited study or not.
Line 241 In Poland, there was a similar trend. In this study, young adults reported the occurrence of at least one somatic symptom 241 (53.5%) and one psychological symptom (72.4%) during the last week.
Is not clear to what “the last week” is referring to.
Line 243 The most intensified proved to be headaches and apathy than any other somatic symptoms like stomach ache, dizziness, nausea or diarrhoea.
This sentence is grammatically not correct in English.
Level 316 As it turned out, the disease was proved by the virus, later named severe acute 316 respiratory syndrome coronavirus 2 (SARS-CoV-2)
As the line above this sentence is not grammatically correct in English
Author Response
Dear Reviewer,
Thank you for your comments, we truly appreciate evevery one of them. Here are our modifications:
- the introduction was extended of the online ways to enhance the mindfulness
- we put the psychological measures before the procedure
- we added more details to the procedure section - hope it is more clear now
- we changed a bit the psychological measures section - we specified the reasons for choosing those questionnaires and extended the definitions of our variables
- the 'participants' section was extended. We added specific information about the recruitment
- in the discussion we specified what are the innovative elements of our research
- we extended the limitations section of the weak spots concerning the procedure, the recruitment process and the ade/sex of participants
- the text was edited by a native speaker so minor grammar mistakes were fixed
Additionally:
- the paragraphs concerning the spread of the COVID in Poland was transfered to the introduction
- the reference concerning the mass media was fixed
- your other comments concerning the grammar and other small mistakes were implemented
Regards,
Aleksandra Patriak
Reviewer 2 Report
Dear authors, thank you very much for your submission to IJERPH and this very current COVID-19 topic! Your paper fits in the journal’s scope, is novel, interesting, and relevant.
However, I have the following issues:
- The paper needs a proofreading by a native speaker. Even if there are not many mistakes, some formulations sound a little odd.
- You should not mention the hypotheses in the introduction but in chapter 2.
- The high number of women in the sample is striking. You should briefly explain this.
- Additionally, the explanation of your examined variables is to shot. You should provide proper definitions with references.
- The limitation regarding the respondents’ age is mentioned too shortly. You should add why this might be a problem. Might older people be better/worse regarding mindfulness, depression, etc.?
Good luck with your revision!
Author Response
Dear Reviewer,
Thank you for your comments, we truly appreciate every one of them. Here are our modifications:
- the introduction was extended of the online ways to enhance the mindfulness
- we put the psychological measures before the procedure
- we added more details to the procedure section - hope it is more clear now
- we changed a bit the psychological measures section - we specified the reasons for choosing those questionnaires and extended the definitions of our variables
- the 'participants' section was extended. We added specific information about the recruitment
- in the discussion we specified what are the innovative elements of our research
- we extended the limitations section of the weak spots concerning the procedure, the recruitment process and the age/sex of the participants
- the text was edited by a native speaker so minor grammar mistakes were fixed
Additionally, we decided to leave the hypothesis in the introduction. We consulted this matter with other scientists and reached the conclusions it should stay as it is. If other reviewers will address this matter we will certainly change it.
Regards,
Authors
Reviewer 3 Report
The article deals with a very interesting topic investigating the associations between anxiety and general psychosomatic functioning during COVID-19 pandemic and analyzing the impact of psychological flexibility and mindfulness.
A particular strength of this Research Article is that it is properly structured and well developed and although the study has some limitations, the authors properly underline them and the research is very interesting.
COVID-19 pandemic triggers some extensive scientific literature about its mental health impacts and several studies investigate the benefits of psychological flexibility and mindfulness in the context of COVID-19 adversity. I suggest that the authors better delineate the state of art of the researches in this field and explain what are the innovative elements of their research.
Just some examples of similar studies:
- Conversano C, Di Giuseppe M, Miccoli M, Ciacchini R, Gemignani A, Orrù G. Mindfulness, Age and Gender as Protective Factors Against Psychological Distress During COVID-19 Pandemic. Front Psychol. 2020 Sep 11;11:1900. doi: 10.3389/fpsyg.2020.01900. PMID: 33013503; PMCID: PMC7516078.
- Matiz A, Fabbro F, Paschetto A, Cantone D, Paolone AR, Crescentini C. Positive Impact of Mindfulness Meditation on Mental Health of Female Teachers during the COVID-19 Outbreak in Italy. Int J Environ Res Public Health. 2020 Sep 4;17(18):6450. doi: 10.3390/ijerph17186450. PMID: 32899739; PMCID: PMC7559290.
- Coyne LW, Gould ER, Grimaldi M, Wilson KG, Baffuto G, Biglan A. First Things First: Parent Psychological Flexibility and Self-Compassion During COVID-19. Behav Anal Pract. 2020 May 6:1-7. doi: 10.1007/s40617-020-00435-w. Epub ahead of print. PMID: 32377315; PMCID: PMC7200171.
- Kroska EB, Roche AI, Adamowicz JL, Stegall MS. Psychological flexibility in the context of COVID-19 adversity: Associations with distress. J Contextual Behav Sci. 2020 Oct;18:28-33. doi: 10.1016/j.jcbs.2020.07.011. Epub 2020 Aug 6. PMID: 32837889; PMCID: PMC7406424.
- Behan C. The benefits of meditation and mindfulness practices during times of crisis such as COVID-19. Ir J Psychol Med. 2020 May 14:1-3. doi: 10.1017/ipm.2020.38. Epub ahead of print. PMID: 32406348; PMCID: PMC7287297.
Author Response
Dear Reviewer,
Thank you for your comments, we truly appreciate every one of them. Here are our modifications:
- the introduction was extended of the online ways to enhance the mindfulness
- we put the psychological measures before the procedure
- we added more details to the procedure section - hope it is more clear now
- we changed a bit the psychological measures section - we specified the reasons for choosing those questionnaires and extended the definitions of our variables
- the 'participants' section was extended. We added specific information about the recruitment
- in the discussion we specified what are the innovative elements of our research
- we extended the limitations section of the weak spots concerning the procedure, the recruitment process and the age/sex of the participants
- the text was edited by a native speaker so minor grammar mistakes were fixed
Regards,
Authors
Reviewer 4 Report
Congratulations to the authors for such an original article. In my opinion, the aim of the study is very interesting and of great relevance.
To improve the paper I suggest developing the following improvements:
- In the introduction I would appreciate finding some references about the evidence about the ability to enhance the level of mindfulness or awareness. There are plenty of papers regarding this issue.
- In the "participants and procedure" it's a lack of explanation about the way to reach the sample. How did the authors reach the study population? How many people did they ask to participate? What is the rate of response?
- In "Psychological measures", when the authors describe the measures used in the study, I'd like to read the reasons for choosing these tools over others.
Author Response

(The authors gave the same response as above.)
